# Lack of Salivary Long Non-Coding RNA XIST Expression Is Associated with Increased Risk of Oral Squamous Cell Carcinoma: A Cross-Sectional Study

**DOI:** 10.3390/jcm10194622

**Published:** 2021-10-08

**Authors:** Tzong-Ming Shieh, Chung-Ji Liu, Shih-Min Hsia, Valendriyani Ningrum, Chiu-Chu Liao, Wan-Chen Lan, Yin-Hwa Shih

**Affiliations:** 1School of Dentistry, China Medical University, Taichung 40402, Taiwan; tmshieh@mail.cmu.edu.tw; 2Department of Oral and Maxillofacial Surgery, MacKay Memorial Hospital, Taipei 104217, Taiwan; cjliu@mmh.org.tw; 3School of Nutrition and Health Sciences, Taipei Medical University, Taipei 110301, Taiwan; bryanhsia@tmu.edu.tw; 4Graduate Institute of Metabolism and Obesity Sciences, Taipei Medical University, Taipei 110301, Taiwan; 5School of Dentistry, Baiturrahmah University, by Pass km 15 Aie Pacah, Padang 25586, West Sumatra, Indonesia; valend888@gmail.com; 6Department of Healthcare Administration, Asia University, Taichung 41354, Taiwan; agi-3@yahoo.com.tw (C.-C.L.); magic1986713@hotmail.com (W.-C.L.)

**Keywords:** long non-coding RNA XIST, oral squamous cell carcinoma, salivary biomarker, morbidity rate

## Abstract

Studies have shown that there is a disparity between males and females in south-east Asia with regard to oral cancer morbidity. A previous study found that oral cancer tissue showed loss of heterozygosity of the X-linked lncRNA XIST gene. We suggest that XIST may play an important role in oral cancer morbidity when associated with sex. Saliva contains proteins and RNAs that are potential biomarkers for the diagnosis of diseases. This study investigated salivary XIST expression and the correlation to clinical–pathological data among oral squamous cell carcinoma patients. Salivary XIST expression was only observed in females, and a high proportion of females with OSCC lack salivary lncRNA XIST expression (88%). The expression showed no correlation with alcohol consumption, betel quid chewing, or cigarette smoking habits. People lacking salivary lncRNA XIST expression had a significantly increased odds ratio of suffering from OSCC (OR = 19.556, *p* < 0.001), particularly females (OR = 33.733, *p* < 0.001). The ROC curve showed that salivary lncRNA XIST expression has acceptable discrimination accuracy to predict the risk of OSCC (AUC = 0.73, *p* < 0.01). Lack of salivary lncRNA XIST expression was associated with an increased risk of OSCC. We provided an insight into the role of salivary lncRNA XIST as a biomarker to predict the morbidity of OSCC.

## 1. Introduction

According to global statistics published by the World Health Organization, oral cavity cancer is among the most prevalent types of cancer worldwide, with the female to male incidence ratio showing a discrepancy of 2:1 in south-east Asia [1]. Excessive alcohol consumption, betel quid chewing, and cigarette smoking (ABC habits) are risk factors for oral cancer [2]. However, the ABC habits cannot explain the increasing trend of young females diagnosed with oral squamous cell carcinoma (OSCC) without performing the ABC habits [3]. 

The long non-coding RNA XIST is an X-linked gene that contributes to X-chromosome inactivation. It is also related to tumorigenesis and progression in nasopharyngeal carcinoma [4], small intestinal adenocarcinoma [5], and breast cancer [6]. A previous study revealed that a loss of genomic copy number variants of XIST is shown in the OSCC group [7]. Recently, one research article provided evidence of a relationship between XIST and the inhibition of tumor progression in vitro [8]. We suggest that this X-linked gene may be associated with the pathogenesis of oral cancer. 

Oral cancer is the sixth leading cause of death in Taiwan in 2020, and the five-year age-standardized relative survival rate is approximately 80% and 38% in stage I and IV, respectively [9]. Early detection and intervention for oral cancer are the main strategies to cease cancer progression and elevate the survival rate. Saliva contains a wide variety of RNA types, and more than 4000 distinct coding or non-coding RNA molecules have been identified [10], some of which are biomarkers of oral cancer [11,12,13,14]. This study aims to identify whether salivary lncRNA XIST is associated with OSCC pathogenesis, and whether it could be a salivary biomarker of OSCC.

## 2. Materials and Methods

### 2.1. Eligibility Criteria

The ethical approval of this study was reviewed and approved by the institutional review board of the MacKay Memorial Hospital, Taiwan (No. 19MMHIS088e). All of the salivary specimens from OSCC patients and individuals who did not suffer from OSCC were collected in previous studies, and stored in the Department of Medical Research, MacKay Memorial Hospital, Tamsui, Taiwan. The saliva was collected by the spitting method, and handled as described previously [15]. Informed consent was obtained from all subjects. Patient and non-patient subjects’ specimens were excluded if GAPDH could not be detected by qPCR. 

### 2.2. Salivary RNA Extraction

Salivary RNA was extracted from 200 µL of salivary specimens with the PureLink RNA kit (Thermo Fisher Scientific, Waltham, MA, USA). The kit yielded 200 µL RNA. We concentrated the sample to 10 µL through ethanol precipitation. The RNA samples were stored at −80 °C until analysis.

### 2.3. RT-qPCR

We used the ReverAid RT reverse transcription kit (Thermo Fisher Scientific, Waltham, MA, USA) and random primers to reverse-transcribe the RNA to cDNA. The primers used for qPCR were designed using the Roche Universal Probe Library (Roche, Basel, Switzerland). qPCR was performed using the Roche LC-480 instrument (Roche, Basel, Switzerland). The reaction conditions were as follows: denaturation at 95 °C for 10 min annealing at 60 °C for 30 s, extension at 72 °C for 1 s, and for a total run of 60 cycles. For qPCR of GAPDH, we used probe no. 45, and for XIST, we used probe no. 32. The primer sequences were as follows: GAPDH forward: GAGTCCACTGGCGTCTTCAC; GAPDH reverse: GTTCACACCCATGACGAACA. XIST forward: TCGGAGAAGGATGTCAAAAGA; XIST reverse: TGCAGCGTGGTATCTTCAAT. Electrophoresis of the amplicons was performed using 4% agarose gels at 100 V.

### 2.4. Statistical Analysis

Statistical analysis of clinical data was conducted using SPSS version 22 software (SPSS Inc., Chicago, IL, USA). The correlation among clinical–pathological data and XIST expression was analyzed by contingency tables, and the significant differences were calculated by Fisher’s exact test, with the correlation shown in Phi value. Dummy variables were replaced with categorical variables to conduct binomial regression analysis and ROC curve analysis. The significant differences between groups were defined when *p* < 0.05. 

## 3. Results

### 3.1. The Characteristics of Participants

Among the 102 participants, 59 were patients with OSCC (male *n* = 33, female *n* = 26) and 43 were individuals without OSCC (the control group) (male *n* = 16, female *n* = 27). The average ages of male and female patients were 53.9 (2.2) and 58.2 (2.3) years old, respectively. The average ages of male and female individuals in the control group were 49.7(2.5) and 39.1(1.3) years old, respectively. Salivary lncRNA XIST was only expressed in females. Among the OSCC group, 35.6% consumed alcohol, 40.7% had a betel nut chewing habit, and 52.5% smoked cigarettes. For primary tumors, 47.5% of cases were T1-T2, and 52.5% were T3-T4. Additionally, 50.8%, 40.7%, and 8.5% of tumors were well, moderately, and poorly differentiated, respectively. For clinical stages, 35.6% of cases were I-II, and 64.4% were III-IV. Only two patients (3.4%) had distant metastasis. No patients showed tumor recurrence. The tumor sites involved were 28.8% buccal, 33.9% tongue, and 37.3% others, including gingiva, floor of the mouth, mandible, and palate (Table 1). 

### 3.2. Salivary lncRNA XIST Was Expressed Only in Females

We conducted a preliminary test to detect XIST expression in buccal cells and saliva, samples of which were kindly provided by four healthy research assistants (two males and two females) in our lab. Of the volunteers, two males and one female did not express XIST in the buccal cells or in the saliva (data shown in Appendix A). We further detected XIST expression in salivary specimens collected from OSCC and from individuals in the control group. Salivary lncRNA XIST was only expressed in females, with a high proportion observed in control group females (Table 1, Figure 1). Control group and OSCC males lacked salivary XIST expression with detectable GAPDH amplicons (data shown in Appendix A). 

The grouping gels, which were cropped from different part of the same gel, or from different gels, were shown with a space. The original full-length gels were included in the Appendix A during peer review process.

### 3.3. Clinical–Pathological Data Difference between Sex among Patients with OSCC 

Among the patients with OSCC, 83% (20 of 24) of the smokers, 90.3% (28 of 31) of those who consumed alcohol, and 95% (20 of 21) of those chewed betel nuts were male. Tumors of male patients were low-grade or well differentiated in 66% (22 of 33) of cases, and most were in the buccal site (13 of 33). A higher proportion of tumors in female patients showed moderate or poor differentiation (17 of 26), and most were on the tongue (14 of 26) (Table 2). Most females with OSCC did not have ABC habits. The tumor was typically small and poorly differentiated when located in the tongue. Most males with OSCC had ABC habits, and the tumors were typically located in the buccal site, were larger, and well differentiated.

### 3.4. Increased Risk of OSCC in Individuals without Salivary lncRNA XIST Expression

We analyzed the correlation between the clinical–pathological data and XIST expression. Salivary lncRNA XIST expression was correlated with sex (Table 1 and Table 3) among all participants, and was correlated with OSCC among female participants (Table 4). Salivary lncRNA XIST expression had no significant correlation with ABC habits or death. We further conducted binomial logistic regression, and found that individuals who did not express XIST had a 19.5-fold higher risk of suffering from OSCC. Females who did not express salivary lncRNA XIST had a 33.7-fold higher risk of suffering from OSCC (Table 5). The ROC analysis showed that, 73% (acceptable discrimination) of the time, the model would correctly assign a higher absolute OSCC risk to patient with an absence of XIST expression (Figure 2).

A patient who lacks salivary XIST expression will have a higher predicted OSCC risk score than a patient with salivary XIST expression. The model will correctly assign a higher absolute OSCC risk to a patient with an absence of XIST expression 73% (acceptable discrimination) of the time.

## 4. Discussion

XIST is a non-coding RNA on the X-chromosome of placental mammals and has a major effect on X-chromosome inactivation. X-chromosome inactivation occurs in males during embryonic development, at the late stage of the first meiotic prophase. However, X-chromosome inactivation persists for the life span of female individuals. One X-chromosome in all female cells undergoes transcriptional inactivation to compensate for the difference in chromosome dosage between sexes [16]. Based on this theory, we evaluated salivary lncRNA XIST expression in all females. Unexpectedly, our data revealed that a large proportion of females with OSCC lacked salivary lncRNA XIST expression (Table 1). These results agree with those of Chang et al. [7], who showed that most patients with oral cancer express lower XIST copy number variants on X-chromosomes. 

The concept of Knudson’s two-hit hypothesis was challenged after the identification of X-linked tumor suppressor genes, in which the single suppression of X-linked tumor suppressor genes caused tumorigenesis. This occurs in several types of tumor, including breast and ovarian cancers [17], sporadic colorectal carcinoma [18], renal cell carcinoma [19], melanoma [20], neuroendocrine tumors [21], and prostate cancer [22]. Regardless of sex, people lacking salivary lncRNA XIST expression had a 19.55-fold increased risk of suffering from OSCC. Females lacking salivary lncRNA XIST had a 33.73-fold increased risk of suffering from OSCC (Table 5). We assumed that the lncRNA XIST played a tumor suppressor role in OSCC, due to a lack of expression increasing the morbidity. However, the actual role of lncRNA XIST in oral cancer still needs to be verified by future in vitro and in vivo experiences.

Oral cancer is the sixth leading cancer in Taiwan, with a high prevalence rate in males. Usually, the primary strategy for preventing OSCC is avoiding risk factors such as ABC habits. The secondary prevention strategy is encouraging people who have been exposed to risk factors to submit to an oral mucosa examination. However, there is an increasing trend for young women diagnosed with OSCC to show no correlation with ABC risk factors [3]. Their oral health condition is ignored and neglected under the criteria of secondary prevention, causing them to delay seeking medical attention. Although the OSCC morbidity rate is lower in females [23], and is not among the top ten leading causes of cancer-related death in Taiwanese women, public health should still pay attention to this small target group. Our study provided salivary lncRNA XIST as biomarker for secondary prevention for females, similar to risk-stratified cancer screening [24] for breast cancer [25,26,27,28,29]. Research indicates that the HPV virus infection is a risk factor of OSCC among Taiwanese females [30]. Early prediction, prevention, and regular examination strategies for high-risk females can reduce the negative effects of the disease, and result in a better prognosis, survival rate, and quality of life.

This study has several limitations. This was a cross-sectional study design. We enrolled participants and collected salivary samples in an oral surgery clinic as much as possible. Therefore, the average age of the OSCC patient between males and females cannot be well controlled. All the participants in this study were Taiwanese, and the outcome cannot represent all OSCC cases around the world.

## 5. Conclusions

A lack of salivary lncRNA XIST expression is associated with an increased risk of OSCC. ROC analysis reveals that salivary lncRNA XIST expression is an acceptable predictor of the risk of developing OSCC.

## Figures and Tables

**Figure 1 jcm-10-04622-f001:**
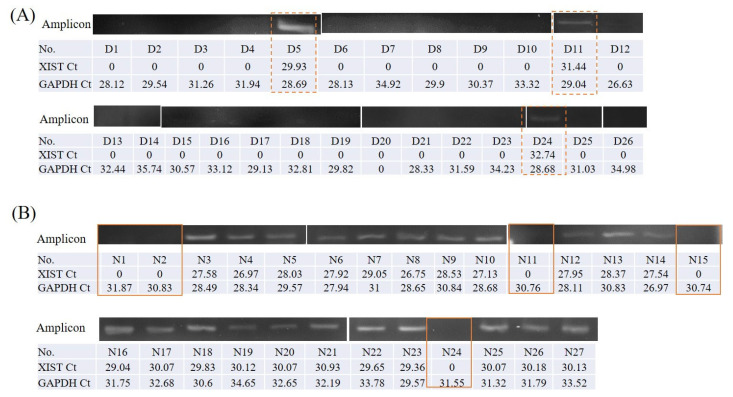
The salivary lncRNA XIST expression in female participants: (**A**) the amplicons and Ct value of XIST and GAPDH among females with OSCC (*n* = 26). The dotted line circles the subjects who express salivary XIST. Only 3 females with OSCC showed positive expression. (**B**) The amplicons and Ct value of XIST and GAPDH among females without OSCC (*n* = 27). The solid line circles the subject who lacks salivary XIST amplicons. Five control group females showing negative expression.

**Figure 2 jcm-10-04622-f002:**
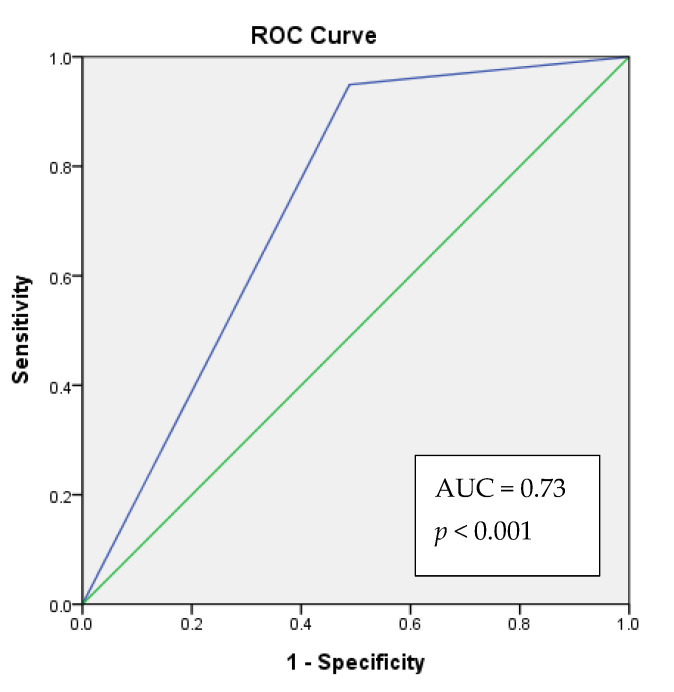
The receiver operating characteristic (ROC) curve analysis of the lack of salivary XIST expression to morbidity prediction of OSCC. Blue line: XIST expression. Green line: reference.

**Table 1 jcm-10-04622-t001:** Characteristics of 102 participants.

	OSCC *n* = 59	Control *n* = 43
Average age, y (mean ± SD)		
Male	53.9 ± 2.2	49.7 ± 2.5
Female	58.2 ± 2.3	39.1 ± 1.3
Variable	n (%)	n (%)
Sex		
Male	33 (55.9)	16 (37.2)
Female	26 (44.1)	27 (62.8)
Salivary lncRNA XIST expression		
Male	0	0
Female	3 (11%)	22 (81%)
Alcohol drinking		
Yes	21 (35.6)	0 (0)
No	38 (64.4)	43 (100)
Betel nut chewing		
Yes	24 (40.7)	0 (0)
No	35(59.3)	43 (100)
Cigarette smoking		
Yes	31 (52.5)	0 (0)
No	28(47.5)	43 (100)
Primary tumor stage		
T1-T2	28 (47.5)	
T3-T4	31 (52.5)	
Differentiation		
Well	30 (50.8)	
Moderate	24 (40.7)	
Poor	5 (8.5)	
Clinical stage		
I-II	21 (35.6)	
III-IV	38 (64.4)	
Distant metastasis (M)		
Yes	2 (3.4)	
No	57 (96.6)	
Recurrence		
Yes	0 (0)	
No	59 (100)	
Tumor site		
Buccal	17 (28.8)	
Tongue	20 (33.9)	
Others	22 (37.3)	

**Table 2 jcm-10-04622-t002:** The significant difference of clinical characteristics between sexes among OSCC patients.

	Sex		
Variable	Male *n* = 33	Female *n* = 26	*p*
Smoking			
Yes	20	4	<0.001 ***
No	13	22	
Alcohol drinking			
Yes	28	3	<0.001 ***
No	5	23	
Betel nut chewing			
Yes	20	1	<0.001 ***
No	13	25	
Differentiation			
Low grade or well	22	9	0.019 *
moderate or poor	11	17	
Diagnosis			
Tongue Ca.	6	14	0.026 *
Buccal Ca.	13	4	
Gingiva Ca.	7	5	
Others	7	3	

Fisher’s exact test * *p* < 0.05, *** *p* < 0.001.

**Table 3 jcm-10-04622-t003:** The correlation and significant difference between XIST expression and clinical pathological data among OSCC patients.

	Sex		Alcohol		Betel		Cigarette		Death	
XIST expression	F	M	No	Yes	No	Yes	No	Yes	No	Yes
Yes	3	0	3	0	3	0	2	1	3	0
No	23	33	25	31	35	21	33	23	38	18
Fishe’s exact test *p* (two-tailed)	0.08		0.1		0.546		1		0.546	
Phi	0.261 *		0.244		0.172		0.035		0.153	

OSCC (male + female, *n* = 59); F: female, M: male; * *p* < 0.05.

**Table 4 jcm-10-04622-t004:** The XIST expression and the correlation of OSCC among females (*n* = 53).

	OSCC		Alcohol		Betel Nut		Cigarette	
XIST expression	No	Yes	No	Yes	No	Yes	No	Yes
Yes	22	3	25	0	25	0	24	1
No	5	23	25	3	27	1	25	3
Fisher exact test *p* (two-tail)	<0.001		0.238		1		0.613	
Phi	0.7 ***		0.231		0.131		0.127	

*** *p* < 0.001; Control group female, *n* = 27; female with OSCC, *n* = 26.

**Table 5 jcm-10-04622-t005:** Binomial logistic regression of OSCC.

		B	S.E.	*p*	OR
All Participants *n* = 102	XIST expression	2.973	0.667	<0.001	19.556
	constant	−1.992	0.615	0.001	0.136
Female subjects *n* = 53	XIST expression	3.518	0.789	<0.001	33.733
	constant	−1.992	0.615	0.001	0.136

Dependent variable: OSCC (0 = no); XIST expression: (0 = express); OR: odds ratio.

## Data Availability

The data that support the findings of this study are available from the corresponding author, [Shih, Y.-H.], upon reasonable request.

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
