# Peer review of "Lack of Salivary Long Non-Coding RNA XIST Expression Is Associated with Increased Risk of Oral Squamous Cell Carcinoma: A Cross-Sectional Study"

_jcm, 2021, doi:10.3390/jcm10194622_

Round 1

Reviewer 1 Report

The submitted article meets the requirements to be considered publishable. It is fun to read, up-to-date and contributes to the importance of biomarkers in the prediction of oral cancer.

Author Response

Thanks for the comment from reviewers.

Reviewer 2 Report

the manuscript entitled with"lack of salivary long noncoding RNA XIST expression is associated with increased risk of oral squamous cell carcinoma: a cross-sectional study", made an attempt to find a predictor of OSCC in saliva. the specific comments are listed below:

  1. In the supplementary data, for XIST detection, in the first picture, the size of the PCR product is 230nt, while in other pictures, the size of XIST PCR product is 127nt, for the same thing, why the authors used different primers? and why the primer sequence for the 230nt product is not shown in the article?
  2. to support the conclusion that XIST only expressed in female, the data in male should be included, other than just say data not shown.
  3. for the introduction part, more information about the relationship between XIST and OSCC could be added.

Author Response

1. Thanks for the reviewer’s comment. For the first picture, it is preliminary data which we test the XIST expression in buccal cells and all cell lines for a grant application; the primer was designed by ourselves using NCBI blast.

We detected the clinical salivary samples using the primer designed by a biotech company and special for qPCR use. This is the reason why the XIST size in the supplementary original gel picture is different from the others.

2. Thanks for the reviewer’s comment. We provide the supplementary Figure2 of the original Ct value and we had sampled part of the male qPCR product to confirm the lack of XIST expression in male saliva by agarose gel electrophoresis.

3. Thanks for the reviewer’s comment. I have signed one recent reference regarding the relationship between XIST and OSCC in the introduction.

Reviewer 3 Report

Comments to Author

1. Study shows GAPDH expression at very high threshold cycle (Ct) ~ 28. The author needs to provide the details of RNA quality and concentration used to for complementary DNA conversion. RNA quality will play significant role in the qPCR extension. 2. The authors made a valid point of focusing their analysis in women’s saliva samples and claimed lack of lcRNA XIST associated with OSCC progression. The average mean age difference is 20 years between OSCC versus normal, the author needs to confirm the consistency of the data outcome after adjusting with a comparable age match. 3. Smoking is the major risk factor for lung & HN cancer, present study shows no correlation with history of ABC (Alcohol drinking, Betel nut chewing and Cigarette smoking) and OSCC. Authors need to add the other possible risk factors for the OSCC in Taiwan women in the discussion.

Author Response

  1. Thanks for the reviewer’s comment. The clinical sample is hard to get from patients once more. Therefore, we do measure the RNA quality in the first place for two samples after the extraction procedure, the value of A260/280 is 1.9~2, and concentration is low but much enough to conduct an RT-PCR (I have to say sorry for didn’t record it down regarding the detail concentration previously).
  2. Thanks for the reviewer’s comment. We had study limitations, subject collection in clinical is as much as possible and difficult to control their age very well. However, we conducted the binominal regression of age and XIST expression in normal females (age range from 23-54). The outcome shows no significant between age and XIST expression (p = 0.628).

  3. Thanks for the reviewer’s comment. Research indicated the HPV virus is significantly associated with oral cancer in Taiwanese females. I‘ve added the reference in the discussion section.